# Decarboxylative oxidation-enabled consecutive C-C bond cleavage

Ruining Li[1], Ya Dong[1], Shah Nawaz Khan [1], Muhammad Kashif Zaman[1], Junliang Zhou[1], Pannan Miao[1], Lifu Hu[1] & Zhankui Sun [1,2,3] ✉

The selective cleavage of C-C bonds is of fundamental interest because it provides an alternative approach to traditional chemical synthesis, which is focused primarily on building up molecular complexity. However, current C-C cleavage methods provide only limited opportunities. For example, selective $C(sp^3)$-$C(sp^3)$ bond cleavage generally relies on the use of transition-metal to open strained ring systems or iminyl and alkoxy radicals to induce β-fragmentation. Here we show that by merging photoredox catalysis with copper catalysis, we are able to employ α-trisubstituted carboxylic acids as substrates and achieve consecutive C-C bond cleavage, resulting in the scission of the inert β-$CH_2$ group. The key transformation relies on the decarboxylative oxidation process, which could selectively generate in-situ formed alkoxy radicals and trigger consecutive C-C bond cleavage. This complicated yet interesting reaction might help the development of other methods for inert $C(sp^3)$-$C(sp^3)$ bond cleavage.

The art of making and breaking bonds has been the driving forces of innovation for chemists[1-3]. It is also the basis of metabolism, enzymology, and biochemistry as a whole[4]. In this context, the cleavage of C-C bonds is of fundamental interest for chemists because it provides an alternative approach to traditional chemical synthesis, which is focused primarily on building up molecular complexity[5,6]. The oxidative cleavage of alkyl aromatics could also lead to the selective degradation of polystyrene related plastics and generate high-value chemicals from abundant polystyrene wastes[7-9]. Despite many impressive advances in this field, the cleavage of inert $C(sp^3)$-$C(sp^3)$ bonds and their subsequent functionalization is still one of the most sought-after challenges in chemistry[10]. Generally, the known strategies for $C(sp^3)$-$C(sp^3)$ bond cleavage could be divided into two classes. One focuses on the strained ring systems employing transition-metal-catalyzed processes that are triggered by C-C bond activation and β-carbon elimination[11,12]. Another process exploits the chemistry of iminyl and alkoxy radicals because of their abilities to break into an alkyl radical species and an unsaturated fragment through β-fragmentation[13-15]. Recently, the group of Sarpong developed the first homolytic C-C bond cleavage method for

the deconstructive diversification of cyclic amines mediated by a silver salt by breaking one $C(sp^3)$-$C(sp^3)$ bond through an in-situ installed hydroxyl group (Fig. 1)[16,17]. Acids are inexpensive, highly stable, and readily available compounds. Decarboxylation enabled $C(sp^2)$-$C(sp^3)$ and $C(sp^2)$-$C(sp^2)$ bond cleavage has been employed for many useful transformations[18-24]. However, little studies have been performed using acids as potential substrates for $C(sp^3)$-$C(sp^3)$ bond cleavage.

We propose to generate peroxyl radicals through the decarboxylative oxidation process[25-29]. We anticipate the photoredox decarboxylation of α-trisubstituted acid **I** would produce tertiary radical **II** which could trap oxygen to create the in-situ formed peroxyl radical **III**. This peroxyl radical then collapses to deliver the radical species **IV** through fragmentation. A few more oxidative transformations from **IV** will provide the final product **VI**.

Here we show, by merging photoredox catalysis with copper catalysis, we are able to use α-trisubstituted acids as substrates and employ a decarboxylative oxidation process to achieve consecutive C-C bond cleavage, resulting in the complete scission of the inert β-$CH_2$ group.

[1]Shanghai Key Laboratory for Molecular Engineering of Chiral Drugs, Pharm-X Center, School of Pharmacy, Shanghai Jiao Tong University, No. 800 Dongchuan Rd., Shanghai 200240, China. [2]Zhangjiang Institute for Advanced Study, Shanghai Jiao Tong University, Shanghai, China. [3]Shanghai Artificial Intelligence Laboratory, Shanghai 200232, China. ✉e-mail: zksun@sjtu.edu.cn

**Fig. 1 | C-C bond cleavage of cyclic amines and proposed reaction design. a** Previous work by Sarpong. **b** This work. **c** Proposed reaction design.

## Results

We began our investigation by using acid **1a** as the substrate and examined different conditions, as shown in Table 1. First we tried **Ir-1** as the photocatalyst and there was little product observed (Table 1, Entry 1). When Cu(OAc)$_2$ was added, we were able to isolate the desired product **1b** with 26% yield (Table 1, Entry 2). The yield was further improved to 48% when ligand **L1** was used (Table 1, Entry 3). A further screening of other ligands revealed **L4** as a better choice (Table 1, Entries 3–8). We then evaluated different copper sources (Table 1, Entries 9–11). When the reaction was carried out at 30 °C, the yield jumped to 81% after 40 h (Table 1, Entry 12). Given the fact the whole transformation is composed of a few steps, the average yield for each step is impressively high. At last, we tested different photocatalysts and **Ir-1** proved to be the best (Table 1, Entries 12–15). Further control experiments revealed no reactions occurred in the absence of photocatalyst, Cs$_2$CO$_3$, or blue LEDs (for a detailed account of the optimization studies, see Supplementary Figs. 12–19).

With the optimized conditions in hand, we proceeded to investigate the scope of this transformation. We first evaluated this method with different aromatic substituted piperidine-4-carboxylic acids (Fig. 2). *N*-substituted piperidine derivatives bearing *tert*-butoxycarbonyl and benzoyl groups were well tolerated (Fig. 2, substrates **1a-2a**). A diverse range of electron-withdrawing and electron-donating functional groups were entirely compatible and delivered the products smoothly (Fig. 2, substrates **3a-8a**). F, Cl, *t*-Bu, and Ph groups furnished the products in good yields (Fig. 2, substrates **3a, 4a, 7a, 8a**). However, strong electron-withdrawing group such as CF$_3$ and strong electron-donating group such as OMe only gave the product in moderate yields (Fig. 2, substrates **5a** and **6a**). Other aromatic group such as thiophene also worked well (Fig. 2, substrate **9a**). For the indole substrate, the cleavage happened between indole and piperidine (Fig. 2, substrate **10a**). Thus, indoline-2,3-dione (**10b**) and piperidin-4-one (**10c**) were isolated instead.

We further applied this method to all carbon cyclic acids (Fig. 2, substrates **11a-14a**). For cyclobutanecarboxylic acid substrates, 1,4-dicarbonyl compounds were isolated in good yields (Fig. 2, substrates **11a-13a**). When cyclopentanecarboxylic acid was used, 1,5-dicarbonyl

compound **14b** was obtained in 65% yield (Fig. 2, substrates **14a**), along with small amount of **11b**, which was the over-oxidized product.

We then examined aliphatic substituted piperidine-4-carboxylic acids (Fig. 3). For these compounds, **Ir-2** turned out to be the better catalyst. Methyl and ethyl groups provided the products in moderate yields (Fig. 3, substrates **15a** and **16a**). However, for allyl, benzyl and isopropyl substituted substrates, oxidation of these functionalities happened and the main isolated products were 4-piperidinone (Fig. 3, substrates **17a-19a**). Four-membered azetidinone substrate gave α-amino ketone product in 58% yield (Fig. 3, substrate **20a**). For five-membered pyrrolidine-3-carboxylic acid substrate **21a**, β-amino ketone product was isolated in 64% yield. As expected, piperidine-3-carboxylic acids furnished γ-amino ketone products in good yields (Fig. 3, substrate **22a, 23a, 25a**). However, due to the oxidation of the benzyl group, the yield for substrate **24a** was low.

We were pleased to find out that this method worked with acyclic acids (Fig. 4). For 2,2,2-triphenylacetic acid and 2,2-diphenylpentanoic acid, benzophenone was isolated as the main product in moderate yields (Fig. 4, substrates **26a-27a**). However, for 2,2-diphenylpropanoic acid, the main product was acetophenone (Fig. 4, substrates **28a**). Different substituted 2-methyl-2-phenylpropanoic acids also furnished acetophenone type products (Fig. 4, substrates **29a-32a**). Accordingly, substrate **33a** provided propiophenone as the main product in good yield. It's noteworthy that 3-phenylpropanoic acids could also be compatible and benzaldehyde type products were isolated (Fig. 4, substrates **34a-36a**). For 3-phenylbutanoic acid, acetophenone was isolated in 66% yield (Fig. 4, substrate **37a**).

We also evaluated this method with β-hydroxy acids (Fig. 5). Interestingly, the C-C cleavage tended to happen at the α-β position, probably because the radical intermediates were stabilized by the β-hydroxy group. Thus, diketones were usually provided as the main products. For substrates which contain tertiary β-hydroxy groups (Fig. 5, substrates **38a-40a**), the products were isolated in good yields. However, substrates with secondary β-hydroxy groups delivered the products only in moderate yields (Fig. 5, substrates **41a-42a**). For substrate **43a**, the C-C cleavage happened at both sides with about 3:1 ratio.

## Table 1 | Optimization of the reaction conditions[a]

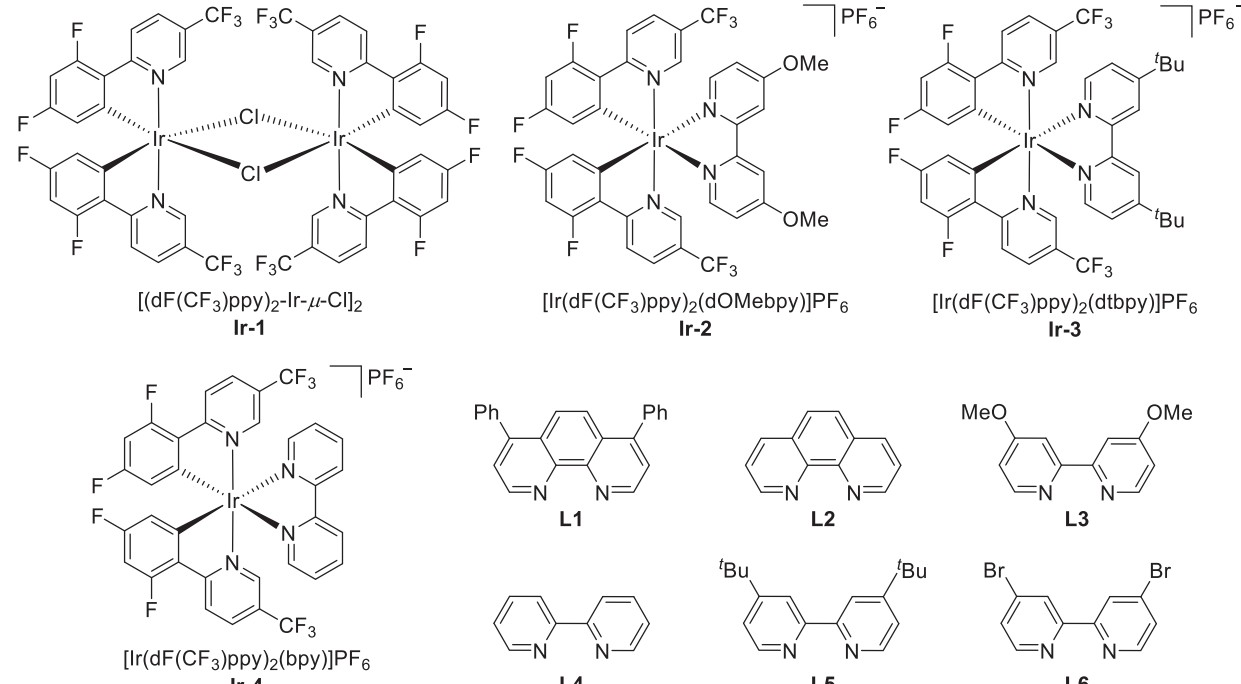

| Entry | Photocatalyst | Copper | Ligand | Yields[b] |
|---|---|---|---|---|
| 1 | **Ir-1** | — | — | <5% |
| 2 | **Ir-1** | Cu(OAc)$_2$ | — | 26% |
| 3 | **Ir-1** | Cu(OAc)$_2$ | **L1** | 48% |
| 4 | **Ir-1** | Cu(OAc)$_2$ | **L2** | 45% |
| 5 | **Ir-1** | Cu(OAc)$_2$ | **L3** | 14% |
| 6 | **Ir-1** | Cu(OAc)$_2$ | **L4** | 49% |
| 7 | **Ir-1** | Cu(OAc)$_2$ | **L5** | 29% |
| 8 | **Ir-1** | Cu(OAc)$_2$ | **L6** | 18% |
| 9 | **Ir-1** | CuO | **L4** | 30% |
| 10 | **Ir-1** | CuBr$_2$ | **L4** | 19% |
| 11 | **Ir-1** | CuCN | **L4** | 6% |
| 12[c] | **Ir-1** | Cu(OAc)$_2$ | **L4** | 81% |
| 13[c] | **Ir-2** | Cu(OAc)$_2$ | **L4** | 48% |
| 14[c] | **Ir-3** | Cu(OAc)$_2$ | **L4** | 60% |
| 15[c] | **Ir-4** | Cu(OAc)$_2$ | **L4** | 50% |

[a]Reaction conditions: **1a** (0.5 mmol), photocatalyst (0.015 mmol), copper (0.1 mmol), ligand (0.125 mmol), Cs$_2$CO$_3$ (0.75 mmol), DCM (10 mL), 45 W blue LEDs, 15 °C. [b]Isolated yield. [c]The reaction was carried out at 30 °C.

**Fig. 2 | Substrate scope of aromatic substituted cyclic acids.** Reaction conditions: substrate (0.5 mmol), **Ir-1** (0.015 mmol), Cu(OAc)$_2$ (0.1 mmol), **L4** (0.125 mmol), Cs$_2$CO$_3$ (0.75 mmol), DCM (10 mL), 45 W blue LEDs, 30 °C, 40 h.

**Fig. 3 | Substrate scope of piperidine-4-carboxylic acids.** Reaction conditions: substrate (0.5 mmol), **Ir-2** (0.015 mmol), Cu(OAc)$_2$ (0.1 mmol), **L4** (0.125 mmol), Cs$_2$CO$_3$ (0.75 mmol), DCM (10 mL), 45 W blue LEDs, 30 °C, 72 h.

**Fig. 4 | Substrate scope of acyclic acids.** Reaction conditions: substrate (0.5 mmol), **Ir-2** (0.015 mmol), Cu(OAc)$_2$ (0.1 mmol), **L4** (0.125 mmol), Cs$_2$CO$_3$ (0.75 mmol), DCM (10 mL), 45 W blue LEDs, 30 °C, 72 h.

**Fig. 5 | Substrate scope of β-hydroxy acids.** Reaction conditions: substrate (0.5 mmol), **Ir-2** (0.015 mmol), Cu(OAc)$_2$ (0.1 mmol), **L4** (0.125 mmol), Cs$_2$CO$_3$ (0.75 mmol), DCM (10 mL), 45 W blue LEDs, 30 °C, 72 h.

Based on our proposed mechanism (Fig. 1), radical intermediate **IV** was formed during the process. To capture this intermediate, we tried to add additives. Eventually, we found the addition of Selectfluor successfully delivered ketone-alcohol as the main product. Thus, under the optimized condition, cyclobutanecarboxylic acid substrates provided 4-hydroxybutyrophenones in good yields (Fig. 6, substrates **44a-46a**). When cyclopentanecarboxylic acid was used, 5-hydroxybutyrophenone was provided instead (Fig. 6, substrates **47a**). For cyclohexanecarboxylic acid, 6-hydroxyhexaphenone was isolated in 75% yield (Fig. 6, substrates **48a**). We also tested other six-member cyclic acids, all of them worked smoothly and delivered the products in moderate to good yields (Fig. 6, substrates **49a-53a**).

This reaction provides a direct method to construct different diketones, which are versatile building blocks in the synthesis of natural products and bioactive compounds. Thus, we performed the synthesis of Primaperone, Melperone and Haloperidol (Fig. 7). These drugs could be accessed in one step from product **12b** via reductive amination in good yields. We also did late-stage modification of commercial drug and complex natural products. For Sertraline derivative **54a**, the reaction worked smoothly and the product was isolated in 83% yield. For steroids **55a** and **56a**, the regioselectivities were good and we only isolated one product. However, the reactions were sluggish and much of the starting materials were recovered.

To better understand the mechanism, we did control experiments (Fig. 8). First, when substrate **15a** was evaluated under the reaction condition, compound **15c** was isolated as a byproduct in 31% yield. However, when **15c** was submitted to the reaction condition, no **15b** was formed, which clearly indicates **15c** is a by-product, not a reacting

intermediate. We did Stern-Volmer experiments and found out the acid **1a** could quench the photocatalyst under basic condition. We also found that no reaction happened when compound **57** was treated with the standard condition. These observations clearly demonstrated that photocatalytic decarboxylation was the initial step and the key to induce the consecutive C-C bond cleavage. We also synthesized aldehyde **58a** and **59a**. When aldehyde **58a** was reacted under standard condition, product **58b** was formed and we observed the formation of formaldehyde[33]. When **59a** were treated with copper, base and O$_2$, we were able to isolate one carbon shorter products **59b**[30].

Based on the above experiments and literatures[29–34], especially the recent publication by Xia[31], we propose a plausible mechanism as shown in Fig. 9. Photocatalyzed decarboxylation of substrate **15a** provides radical **A**, which is captured by oxygen to generate peroxyl radicals **B**. Intermediate **B** might form the dioxetane intermediate **C** through hydrogen atom transfer (HAT). Upon thermal cleavage, it delivers dicarbonyl compound **D**. The oxidative dehomologation of intermediate **D** renders **15b** as the final product. In another possible way, intermediate **E** might form. The extrusion of formaldehyde through β-scission delivers intermediate **F**[33], which is further oxidized by oxygen to provide **15b** as the final product[29].

## Discussion

In conclusion, we have successfully developed consecutive C-C bond cleavage by taking the advantages of photoredox catalysis along with copper catalysis. This complicated process exploits the use of stable α-trisubstituted acids as substrates and efficiently breaks three C-C bonds at the same time. The key transformation features a

**Fig. 6 | Substrate scope for the ketone-alcohol products.** Reaction conditions: substrate (0.5 mmol), **Ir-1** (0.015 mmol), Cu(OAc)$_2$ (0.1 mmol), **L4** (0.125 mmol), Cs$_2$CO$_3$ (0.75 mmol), Selectfluor (0.75 mmol), DCM (10 mL), 45 W blue LEDs, 30 °C, 72 h. Yields were determined by $^1$H NMR.

**a**

**12b**

**85%**

**Primaperone**

**83%**

**Melperone**

**80%**

**Haloperidol**

**54a**

3 mol% **Ir-2**
20 mol% Cu(OAc)₂, 25mol% **L4**

1.5 eq Cs₂CO₃, DCM, O₂ baloon
45W blue LEDs, 30 °C, 72h

**54b**, 83%

**b**

**55a**

3 mol% **Ir-1**
20 mol% Cu(OAc)₂, 25mol% **L4**

1.5 eq Cs₂CO₃, 1.5 eq Selectfluor
DCM, O₂, blue LEDs, 30 °C, 40h

**55b**, 50% (brsm)

**56a**

3 mol% **Ir-1**
20 mol% Cu(OAc)₂, 25mol% **L4**

1.5 eq Cs₂CO₃, 1.5 eq Selectfluor
DCM, O₂, blue LEDs, 30 °C, 40h

**56b**, 35% (brsm)

**Fig. 7 | Synthetic utilities. a** Synthesis of commercial drugs. **b** Modification of natural products.

**a**

**b**

**Fig. 8 | Experimental observations for mechanism studies. a** The model reaction and the main by-product. **b** Control experiments.

decarboxylative oxidation process to generate the in-situ formed alkoxy radical which could trigger fragmentation and the following oxidation process. We believe this finding might shed light on the development of other methods for inert C(sp³)-C(sp³) bond cleavage.

## Methods

**General procedure for consecutive C-C bond cleavage products**
To a 50 ml round bottomed flask equipped with a magnetic stirrer bar were added acid (0.5 mmol, 1.0 equiv.), **Ir-1** or **Ir-2** (0.015 mmol, 0.03 equiv.), Cu(OAc)$_2$ (18.1 mg, 0.1 mmol, 0.2 equiv.), 2,2′-bipyridine (19.5 mg, 0.125 mmol, 0.25 equiv.), Cs$_2$CO$_3$ (245 mg, 0.75 mmol, 1.5 equiv.) and DCM (10 mL). The flask was quickly degassed three times and flushed with oxygen through balloon, and then the mixture was heated to 30 °C in an oil bath and irradiated with three 45 W blue LEDs (5 cm away) for 40 h or 72 h. The reaction mixture was filtered and

concentrated. The residue was purified by column chromatography on silica with petroleum ether/ethyl acetate mixture as the eluent.

**General procedure for ketone-alcohol products**
To a 50 ml round bottomed flask equipped with a magnetic stirrer bar were added acid (0.5 mmol, 1.0 equiv.), **Ir-1** (22.3 mg, 0.015 mmol, 0.03 equiv.), Cu(OAc)$_2$ (18.1 mg, 0.1 mmol, 0.2 equiv.), 2,2′-bipyridine (19.5 mg, 0.125 mmol, 0.25 equiv.), Cs$_2$CO$_3$ (245 mg, 0.75 mmol, 1.5 equiv.), Selectfluor (265 mg, 0.75 mmol, 1.5 equiv.) and DCM (10 mL). The flask was quickly degassed three times and flushed with oxygen through balloon, and then the mixture was heated to 30 °C in an oil bath and irradiated with three 45 W blue LEDs (5 cm away) for 40 h. The reaction mixture was filtered and concentrated. The residue was purified by column chromatography on silica with petroleum ether/ethyl acetate mixture as the eluent.

**Fig. 9 | Proposed mechanism.** The reaction features a decarboxylative oxidation process and might proceed through two different mechanisms.

## Data availability

Materials and methods, optimization studies, experimental procedures, mechanistic studies, $^1$H NMR spectra, $^{13}$C NMR spectra and mass spectrometry data generated in this study are provided in the Supplementary Information file.

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

## Acknowledgements

We thank Shanghai Jiao Tong University for financial support (WF220417003 to Z.S.; AF1700038 to R.L.).

## Author contributions

R.L. and Y.D. contributed equally to this paper. Z.S. conceived the project. R.L., Y.D., S.N.K., M.K.Z., J.Z., P.M. and L.H. performed all experiments. All authors analysed the results. Z.S. and R.L. wrote the manuscript.

## Competing interests
The authors declare no competing interests.
