## [Peer Review File · Nature Communications]

REVIEWER COMMENTS

Reviewer #1 (Remarks to the Author):

This paper describes a consecutive C-C bond cleavage and functionalization of readily available carboxylic acids via photooxidation. Though with limitations, many kinds of carbonyl containing structures can be obtained by this Ir/Cu catalytic system. The cleavage of three different C-C bonds happened smoothly in one pot under visible light irradiation and the reaction could be applied in late-stage modification of complex drugs and natural products, highlighting the synthetic value of this reaction. The novelty is enough. However, there are some problems. I recommend its publication after some major considerations to the following points:

1) Reaction setup. This reaction used three 45 W blue LEDs as light sources. In my experience, sufficient heat dissipation device is needed to cool these LEDs down to room temperature, yet the authors claim that the flask containing reaction mixture was placed in oil bath to keep it at 30 °C. It is necessary to put a picture of the reaction setup in the supporting information for others to reproduce this result, and explain how to control the temperature.

2) Mechanism. The proposed mechanism is problematic. I am confused by the transformation of B to C in Figure 5. How does a peroxy radical give an oxygen-centered radical after one β -fragmentation process? How does the peroxide radical intermediate F transfer to 10b under oxidation conditions? What the real roles of Ir, Cu, ligand and O₂? The authors gave no solid evidence to support the catalytic cycle. Do the authors observe the proposed intermediate $\bullet\text{O}_2^-$, which is highly active? And what happened to it? No reaction cycle provided. Many photocatalysis related mechanism experiments are required to support the proposed catalytic cycle. Radical trapping experiments are needed to prove a radical mechanism. Also, can the authors provide more evidences, such as Stern–Volmer quenching experiment, to support the proposed reductive quenching cycle, which is different from MacMillan's mechanism (ref 26)?

3) Substrate scope. The paper shows that this method is compatible with different arenes including benzene and thiophene. Does this reaction tolerate nitrogen containing heteroarenes, such as pyridine, pyrrole, and indole, since these structures are commonly seen in bioactive molecules and are important in drug synthesis?

4) Role of Selectfluor. In Table 7, Selectfluor was used to capture the radical intermediate IV and to give the ketone-alcohol product. Selectfluor is generally used as an oxidant but it seems to be a

reductive process in this case. Is there any experimental or documentary basis to point out the role of Selectfluor? And why the substrates 54a & 55a lead to aldehyde products in the same condition?

5) Phenyl radical. In Table 5, benzophenone was obtained in the standard condition when 2,2,2-triphenylacetic acid was used. A phenyl radical was likely formed in this transformation based on the proposed mechanism. To my knowledge, it is very rare to see that a phenyl radical is formed in β -fragmentation. Could authors do some experiments to capture the radical or at least monitor the corresponding byproduct?

6) Gram-scale Reaction. Gram-scale reactions should be demonstrated since it is important to evaluate the application potential of a photocatalyzed reaction.

7) Spelling. In Discussion, line 6, "could trig β -fragmentation" should be "could trigger β -fragmentation". In Supporting Information, General Information, line 12, "protio DMSO-d6" should be "protic DMSO-d6". In addition, authors should pay more attention to the space between number and unit, both in text and in reaction formula.

8) In Table 2, how the substituent's electronic properties of the aryl rings influenced the result of this reaction? In addition, when aromatic substituted piperidine-4-carboxylic acids changed to allyl, benzyl and isopropyl substituted piperidine-4-carboxylic acids, 4-piperidinones were isolated as main products. How it happened? The byproducts should be mentioned. Why the substrates 21a-24a do not undergo consecutive C-C bond cleavage or excessive oxidation? And what product will be obtained using 48a as the substrate? What are the byproducts in Table 5?

9) This is a photooxidative reaction using oxygen as the oxidant, but the related references are limited.

Reviewer #2 (Remarks to the Author):

This manuscript reports a decarboxylation triggered oxidative C-C bond cleavage utilizing the dual catalysis of Cu and photocatalyst. Upon the irradiation, this method facilitates the α -trisubstituted carboxylic acids to undergo a radical decarboxylation and trapped by O₂ to generate the peroxide

radical, which is also a key intermediate for cumene and even polystyrene oxidations through the oxidative C(sp³)-C(sp³) bond cleavage (Chin, *J. Chem.* 2021, 39, 3225; *Sci. China Chem.*, 2021, 64, 1487; *ChemSusChem* 2021,14, 5049; *Org. Lett.* 2021, 23, 4057). Alternatively, in this paper, a consecutive two C(sp³)-C(sp³) bonds cleavage results the final dicarbonyl products, which provides significant novelty. A variety of α -trisubstituted carboxylic acids are tolerated well to afford products in good yields. The results in the remote carbonyl alcohols formation in the presence of selectfluoro are also promising. This straightforward method is worthy of publishing, but more work is needed to improve the understanding in the mechanism. Therefore, I recommend for publication in *Nat. Commun.* after the following points have been addressed.

In the proposed mechanism, two equivalents of CO₂ were generated during the reaction. One equiv from the direct decarboxylation and the other from intermediate D, however, the reaction only used 1.5 equiv of base. Please examine the various amounts of the base, or please make a comment in the main text.

Following the question above, the low yield in the reaction of 56a might suggest an alternative process from intermediate C to E, which involves the direct β -scission to generate formaldehyde as byproduct (*J. Am. Chem. Soc.* 2019, 141, 10556). Could the authors examine the GC-MS of the residue solution to see if the formaldehyde is formed. If so, this might explain why 1.5 equiv of base is enough.

The authors argued that the copper catalyst was responsible for the oxidation of intermediate C to aldehyde and then acid. However, when the all carbon cyclic acids were used, the aldehydes instead of the acids or products with shorter chains are formed. Although the copper has indeed used for oxidation of alcohols (*Angew. Chem. Int. Ed.* 2014, 53, 8824), a cocatalyst is typically required. To further support this argument, I would suggest the authors to conduct such oxidations under standard conditions using the alcohol and aldehyde in the absence of the photocatalyst and visible light to see if the aldehyde and acid are formed.

Please also examine the reaction in the absence of copper catalyst solely. How about the reaction using 3 mol% of Ir-1 catalyst and 5 mol% of L4 in the absence of the copper catalyst?

Other minor issues:

From intermediate B to C, I would not say it as a β -fragmentation since two sigma bonds are cleaved and the C-C bond is the gamma-bond to the radical in B.

'baloon' should be balloon.

48% percent should be 48%.

In the ¹H NMR of 24b, we can observe the product 21b, please calculate the corresponding yields based on the NMR spectrum and make a comment in the manuscript.

Large amounts of the grease were observed in the spectra of lines 1557, 1559, 1586, 1590, 1594-1598, and 1602, which might mislead the isolated yields. Please purify the spectra.

cite the references above properly.

Reviewer #1 (Remarks to the Author):

This paper describes a consecutive C-C bond cleavage and functionalization of readily available carboxylic acids via photooxidation. Though with limitations, many kinds of carbonyl containing structures can be obtained by this Ir/Cu catalytic system. The cleavage of three different C-C bonds happened smoothly in one pot under visible light irradiation and the reaction could be applied in late-stage modification of complex drugs and natural products, highlighting the synthetic value of this reaction. The novelty is enough. However, there are some problems. I recommend its publication after some major considerations to the following points:

1) Reaction setup. This reaction used three 45 W blue LEDs as light sources. In my experience, sufficient heat dissipation device is needed to cool these LEDs down to room temperature, yet the authors claim that the flask containing reaction mixture was placed in oil bath to keep it at 30 °C. It is necessary to put a picture of the reaction setup in the supporting information for others to reproduce this result, and explain how to control the temperature.

Thanks for pointing out. Most of the reactions were carried out in winter when the temperature was low and the air conditioning was not working properly about (15°C). So, we used an oil bath to keep the reaction at 30 °C. The pictures were taken and provided in the SI.

2) Mechanism. The proposed mechanism is problematic. I am confused by the transformation of B to C in Figure 5. How does a peroxy radical give an oxygen-centered radical after one β -fragmentation process? How does the peroxide radical intermediate F transfer to 10b under oxidation conditions? What the real roles of Ir, Cu, ligand and O₂? The authors gave no solid evidence to support the catalytic cycle. Do the authors observe the proposed intermediate •O₂-, which is highly active? And what happened to it? No reaction cycle provided. Many photocatalysis related mechanism experiments are required to support the proposed catalytic cycle. Radical trapping experiments are needed to prove a radical mechanism. Also, can the authors provide more evidences, such as Stern–Volmer quenching experiment, to support the proposed reductive quenching cycle, which is different from MacMillan's mechanism (ref 26)?

Thanks for pointing out.

As suggested, we did radical trapping experiments. When TEMPO was added, the reaction was completely quenched.

We also performed Stern–Volmer quenching experiments under different conditions and found out the substrate **1a** and Selectfluor could quench the photocatalyst in the presence of Cs₂CO₃.

We were able to detect the formation of formaldehyde during the reaction (J. Am. Chem. Soc. 2019, 141, 10556). Therefore, we believe the original mechanism we proposed is wrong. Based on the observation and the literature, we propose a new possible mechanism:

3) Substrate scope. The paper shows that this method is compatible with different arenes including benzene and thiophene. Does this reaction tolerate nitrogen containing heteroarenes, such as pyridine, pyrrole, and indole, since these structures are commonly seen in bioactive molecules and are important in drug synthesis?

As suggested, we synthesized the indole substrate and tried it under standard condition. However, we isolated the indoline-2,3-dione and piperidin-4-one. The cleavage happened between indole and piperidine ring

As for pyridine substrate, the reaction didn't proceed under standard condition.

4) Role of Selectfluor. In Table 7, Selectfluor was used to capture the radical intermediate IV and to give the ketone-alcohol product. Selectfluor is generally used as an oxidant but it seems to be a reductive process in this case. Is there any experimental or documentary basis to point out the role of Selectfluor? And why the substrates 54a & 55a lead to aldehyde products in the same condition?

In our original design, we tried to capture the radical intermediate with Selectfluor in order to introduce F to the product. However, it delivered the ketone-alcohol product.

From Stern–Volmer quenching experiments, Selectfluor was able to quench the photocatalyst with Cs₂CO₃. However, without Cs₂CO₃, no liner quenching effect was observed. We also tried other amines such as Tetrabutylammonium iodide, Tetrabutylammonium bromide, NH₄Cl, and TEA

(Figure S19). All of them could deliver the ketone-alcohol product, although in lower yield. From these results, it's possible that Selectfluor was oxidized by the photocatalyst under basic condition and generated an α -aminoalkyl radical which coupled with oxygen-centered radical to give acetal intermediate. Upon hydrolysis, it will deliver the ketone-alcohol product.

For substrates **55a** & **56a**, without Selectfluor, it were messy reactions. The starting material totally disappeared and no obvious products could be isolated. However, with Selectfluor, we were able to isolate aldehyde products. It's possible that the ketone-alcohol product of these steroids could be more easily oxidized to aldehyde under the same condition.

5) Phenyl radical. In Table 5, benzophenone was obtained in the standard condition when 2,2,2-triphenylacetic acid was used. A phenyl radical was likely formed in this transformation based on the proposed mechanism. To my knowledge, it is very rare to see that a phenyl radical is formed in β -fragmentation. Could authors do some experiments to capture the radical or at least monitor the corresponding byproduct?

As suggested, we did the experiment and we observed the byproduct benzoquinone and phenol by GC-MS

6) Gram-scale Reaction. Gram-scale reactions should be demonstrated since it is important to evaluate the application potential of a photocatalyzed reaction.

As suggested, we performed the gram-scale reaction.

7) Spelling. In Discussion, line 6, “could trig β -fragmentation” should be “could trigger β -fragmentation”. In Supporting Information, General Information, line 12, “protio DMSO-d6” should be “protic DMSO-d6”. In addition, authors should pay more attention to the space between number and unit, both in text and in reaction formula.

Thanks for pointing out. We made corrections.

8) In Table 2, how the substituent's electronic properties of the aryl rings influenced the result of this reaction? In addition, when aromatic substituted piperidine-4-carboxylic acids changed to allyl, benzyl and isopropyl substituted piperidine-4-carboxylic acids, 4-piperidinones were isolated as main products. How it happened? The byproducts should be mentioned. Why the substrates 21a-24a do not undergo consecutive C-C bond cleavage or excessive oxidation? And what product will be obtained using 48a as the substrate? What are the byproducts in Table 5?

From Table 2, we could see electron-withdrawing group certainly decrease the efficiency and lower the yield. For example, CF₃ substitution gave only 52% yield. Pyridine substrate didn't reaction at all.

On the other side, strong electron-donating group might lead to the oxidation of the aromatic ring and the yield would be decreased, too. For example, methoxy group only gave 66% yield. Indole substrate gave indoline-2,3-dione product.

For allyl, benzyl and isopropyl substituted piperidine-4-carboxylic acids, the cleavage happened on the other side to form more stable radical intermediate (R). This radical intermediate (R) will be further oxidized just like the indole substrate.

For substrates **21a-24a**, no obvious consecutive C-C bond cleavage product were isolated. However, for **24a** (**25a**), we did observed small amount of **22b**, which was the over-oxidized product. So, we believe for piperidine-4-carboxylic acid substrates, the consecutive cleavage is easier.

9) This is a photooxidative reaction using oxygen as the oxidant, but the related references are limited.

Thanks for pointing out. We have added related references 27-32.

Reviewer #2 (Remarks to the Author):

This manuscript reports a decarboxylation triggered oxidative C-C bond cleavage utilizing the

dual catalysis of Cu and photocatalyst. Upon the irradiation, this method facilitates the α -trisubstituted carboxylic acids to undergo a radical decarboxylation and trapped by O₂ to generate the peroxide radical, which is also a key intermediate for cumene and even polystyrene oxidations through the oxidative C(sp³)-C(sp³) bond cleavage (Chin, *J. Chem.* 2021, 39, 3225; *Sci. China Chem.*, 2021, 64, 1487; *ChemSusChem* 2021,14, 5049; *Org. Lett.* 2021, 23, 4057). Alternatively, in this paper, a consecutive two C(sp³)-C(sp³) bonds cleavage results the final dicarbonyl products, which provides significant novelty. A variety of α -trisubstituted carboxylic acids are tolerated well to afford products in good yields. The results in the remote carbonyl alcohols formation in the presence of selectfluoro are also promising. This straightforward method is worthy of publishing, but more work is needed to improve the understanding in the mechanism. Therefore, I recommend for publication in *Nat. Commun.* after the following points have been addressed.

In the proposed mechanism, two equivalents of CO₂ were generated during the reaction. One equiv from the direct decarboxylation and the other from intermediate D, however, the reaction only used 1.5 equiv of base. Please examine the various amounts of the base, or please make a comment in the main text.

We examined the various amounts of the base, as shown in Figure S15. 1.5 equiv and 2 equiv of bases didn't give too much difference.

We believe the original mechanism we proposed is wrong. Based on the observation and the literature, we propose a new possible mechanism:

Following the question above, the low yield in the reaction of 56a might suggest an alternative process from intermediate C to E, which involves the direct β -scission to generate formaldehyde as byproduct (*J. Am. Chem. Soc.* 2019, 141, 10556). Could the authors examine the GC-MS of the residue solution to see if the formaldehyde is formed. If so, this might explain why 1.5 equiv of base is enough.

Thanks for pointing out. You are right. We redid the reaction of 56a to 56b. We did detect the formation of formaldehyde as byproduct. Therefore we changed our mechanism, as above mentioned. Thanks again for pointing out.

The authors argued that the copper catalyst was responsible for the oxidation of intermediate C to aldehyde and then acid. However, when the all carbon cyclic acids were used, the aldehydes instead of the acids or products with shorter chains are formed. Although the copper has indeed used for oxidation of alcohols (Angew. Chem. Int. Ed. 2014, 53, 8824), a cocatalyst is typically required. To further support this argument, I would suggest the authors to conduct such oxidations under standard conditions using the alcohol and aldehyde in the absence of the photocatalyst and visible light to see if the aldehyde and acid are formed.

Thanks for the suggestions. We tried the following reaction without photocatalyst and visible light. The conversion of **59a** to **59b** is very low. So, copper alone is not good enough to convert alcohols into aldehyde.

We also tried the reaction of **60a**. This time, we isolated **60b** in 50% yield.

Please also examine the reaction in the absence of copper catalyst solely. How about the reaction using 3 mol% of Ir-1 catalyst and 5 mol% of L4 in the absence of the copper catalyst?

We tried the condition suggested in **Figure S18**. Using 3 mol% of Ir-1 catalyst and 5 mol% of L4 in the absence of the copper catalyst didn't give the desired product.

Other minor issues:

From intermediate B to C, I would not say it as a β -fragmentation since two sigma bonds are cleaved and the C-C bond is the gamma-bond to the radical in B.

Thanks for the suggestions. We made changes accordingly.

'baloon' should be balloon.

Thanks for the suggestions. We made changes accordingly.

48% percent should be 48%.

Thanks for the suggestions. We made changes accordingly.

In the ^1H NMR of 24b, we can observe the product 21b, please calculate the corresponding yields based on the NMR spectrum and make a comment in the manuscript.

Thanks for pointing out. We changed the yield and made the comment.

Large amounts of the grease were observed in the spectra of lines 1557, 1559, 1586, 1590, 1594-1598, and 1602, which might mislead the isolated yields. Please purify the spectra.

Thanks for pointing out. These products are primary alcohol and very polar. They came out with the bpy ligand. So, we had problem isolating clean products. Therefore, we changed to HNMR yield.

cite the references above properly.

Thanks for pointing out. We cited new references 27-32

REVIEWER COMMENTS

Reviewer #1 (Remarks to the Author):

The authors have addressed some of the comments, but some reversion are still required.

- 1) Reaction setup. The picture is not informative and no light source is shown.
- 2) Mechanism. I believe the revised mechanism is still problematic since intermediate B and the other similar intermediate are hardly formed in situ during the reaction. And the role of the copper catalyst is still missing. Recently, the Xia group reported a similar reaction catalyzed by iron (Green Chem., 2022, 24, 5553-5558). The mechanism they proposed is more convincing.
- 3) Stern-Volmer luminescence quenching experiments. As shown in Figure S21-34, there are some fitting straight lines, but the slope of the straight line seems wrong. For example, in Figure S23 or S27, based on Linear Least Squares, the slope of the fitting line should be closer to the red line in the figure below. And the R2 is too low. Please check the results and show reasonable charts, the charts are not acceptable.
- 4) On page S27, to detect phenol and 1,4-benzoquinone, the MS spectra have been shown. Can the author show the total GC-MS spectra?

Reviewer #2 (Remarks to the Author):

The manuscript has been improved significantly from the previous version after addressing the comments from the reviewers.

Still some concerns on the mechanism, in particular, a biggest shortcoming of the current paper, in this reviewer's opinion, is that the authors barely discussed the results of the mechanistic studies while merely pasted the figures in the SI.

Here are some specific points that needs to be remedied.

- 1) The Stern-Volmer experiments were conducted, and a conclusion that involved "the acid 11a could quench the photocatalyst" was given in the main text. A few figures have been pasted in the SI, however, the figures are not organized well and no details had been discussed at all in the main

text or the SI, which makes it difficult for the readers to follow the claims and verify them from the data. The descriptions as well as the corresponding references to the data/figures are highly recommended to be added.

2) The result on the observation of 10d, 10e, and 10f was removed from the main text, while the reaction was still described in the SI. I would recommend to remain the description in the main text if the editors agree.

3) The results on the mechanistic studies are highly recommended to be reorganized in the SI. At least some subtitles and reaction details could be added to lead the readers to follow the design logic.

4) This reviewer appreciated the result on the formaldehyde capturing. This result was crucial to prove that the radical scission instead of the oxidation happened during the reaction to cause the second C-C bond cleavage. However, such a result was only discussed in the response letter instead of in the main text. Please discuss it in the main text.

5) A new mechanism involving the formation of peroxide was proposed, see fig. 5 in the current paper. Do the author have any evidences to support such an intermediate? If not, please delete it. The generation of D is no doubt, however in this reviewer's opinion, the intermediate A could react with O₂ to generate D through either a peroxide radical (int B in the previous vision) or a peroxide (int no name in the current vision). If the authors are not sure about it, do not draw it in fig. 5.

6) the results on the C(sp³)-C(sp³) bond cleavage in ref. 27-29 should be discussed in the introduction.

Reviewer #1 (Remarks to the Author):

The authors have addressed some of the comments, but some reversions are still required.

1) Reaction setup. The picture is not informative and no light source is shown.

The information about the LEDs was provided in General Information on Page S3:

The blue LEDs were purchased from supermarket and directly used without any filters. The brand name of LEDs is Jin Dian Yuan and the model number is JDY-TG01.

The pictures were provided in General Procedures and Experimental Data on Page 36.

2) Mechanism. I believe the revised mechanism is still problematic since intermediate B and the other similar intermediate are hardly formed in situ during the reaction. And the role of the copper catalyst is still missing. Recently, the Xia group reported a similar reaction catalyzed by iron (Green Chem., 2022, 24, 5553-5558). The mechanism they proposed is more convincing.

Thanks for providing the information. That's very helpful. We also did control experiments which clarify the role of copper catalyst. It's responsible for the dehomologation reaction.

Based on Green Chem. (2022, 24, 5553-5558), ChemSusChem (2016, 9, 241-245), our observations and your suggestions, I believe the following mechanism is the most reasonable one:

However, as we observed the formation of formaldehyde, the other reviewer suggested the formation of intermediate F from intermediate B, though either a peroxide radical or a peroxide. I feel it's hard to refuse his suggestion. So, the possibility of another mechanism could not be ruled out (Isr. J. Chem. 60, 410-415 (2020); J. Am. Chem. Soc. 141, 10556-10564 (2019)).

So, in the main text, we combined these two mechanisms together.

3) Stern-Volmer luminescence quenching experiments. As shown in Figure S21-34, there are some fitting straight lines, but the slope of the straight line seems wrong. For example, in Figure S23 or S27, based on Linear Least Squares, the slope of the fitting line should be closer to the red line in the figure below. And the R2 is too low. Please check the results and show reasonable charts, the charts are not acceptable.

Thanks for pointing out. We redid the Stern-Volmer experiments. Now the fitting is better and R2 is high. Please see page

4) On page S27, to detect phenol and 1,4-benzoquinone, the MS spectra have been shown. Can the author show the total GC-MS spectra?

The full MS spectra is shown on page S28

Reviewer #2 (Remarks to the Author):

The manuscript has been improved significantly from the previous version after addressing the comments from the reviewers.

Still some concerns on the mechanism, in particular, a biggest shortcoming of the current paper, in this reviewer's opinion, is that the authors barely discussed the results of the mechanistic studies while merely pasted the figures in the SI.

Here are some specific points that needs to be remedied.

1) The Stern-Volmer experiments were conducted, and a conclusion that involved “the acid 11a could quench the photocatalyst” was given in the main text. A few figures have been pasted in the SI, however, the figures are not organized well and no details had been discussed at all in the main text or the SI, which makes it difficult for the readers to follow the claims and verify them from the data. The descriptions as well as the corresponding references to the data/figures are highly recommended to be added.

Thanks for pointing out. We reorganized these materials in the main text and the SI. Please see page S26-S35.

The other reviewer suggested to cite the recent publication of Xia (Green Chem. 2022, 24, 5553-5558), and the mechanism proposed in his studies:

So, I think the reaction is kind of similar. Based on the suggestion of the other reviewer, the following mechanism could not be ruled out:

So, it's possible that the reaction proceeds under two ways. So, we proposed both mechanisms in the main text and the SI.

2) The result on the observation of 10d, 10e, and 10f was removed from the main text, while the reaction was still described in the SI. I would recommend to remain the description in the main text if the editors agree.

Thanks for pointing out. Based on your and the other reviewer's suggestion, we have modified the mechanism in the main text and the SI.

3) The results on the mechanistic studies are highly recommended to be reorganized in the SI. At least some subtitles and reaction details could be added to lead the readers to follow the design logic.

Yes, we have reorganized the mechanistic studies in the SI. Please see page S26-S35.

4) This reviewer appreciated the result on the formaldehyde capturing. This result was crucial to prove that the radical scission instead of the oxidation happened during the reaction to cause the second C-C bond cleavage. However, such a result was only discussed in the response letter instead of in the main text. Please discuss it in the main text.

Thanks for pointing out. We discussed in the main text. Please see P17.

5) A new mechanism involving the formation of peroxide was proposed, see fig. 5 in the current paper. Do the author have any evidences to support such an intermediate? If not, please delete it. The generation of D is no doubt, however in this reviewer's opinion, the intermediate A could react with O₂ to generate D though either a peroxide radical (int B in the previous vision) or a peroxide (int no name in the current vision). If the authors are not sure about it, do not draw it in fig. 5.

Thanks for pointing out. We have modified the mechanism.

6) the results on the C(sp³)-C(sp³) bond cleavage in ref. 27-29 should be discussed in the introduction.

Yes, ref. 27-29 focused on the cleavage of alkyl aromatics. So, we have discussed it in the introduction.

REVIEWERS' COMMENTS

Reviewer #1 (Remarks to the Author):

The revised manuscript is suitable for publication.

Reviewer #2 (Remarks to the Author):

The authors have improved the manuscript from the previous version. The work may be accepted for publication in Nat. Commun.